# Evaluating the longitudinal physical and psychological health effects of persistent long Covid 3.5 years after infection

Gregory Vallée[1], David Xi [ID][1], Gordana Avramovic[2], Brendan O'Kelly[3], John S. Lambert [ID][1,2]*

1 Catherine McAuley Education & Research Centre, University College Dublin, Dublin, Ireland,
2 Department of Infectious Diseases, Catherine McAuley Education & Research Centre, Mater Misericordiae University Hospital, Dublin, Ireland, 3 Department of Infectious Diseases, Our Lady of Lourdes Drogheda, Beaumont Hospital, Dublin, Ireland

* jlambert@mater.ie

## Abstract

This is a 3.5-year single-center observational cohort study investigating the longitudinal impact of Long Covid on the physical and mental health of patients. Patients were assessed at 3 months, 1 year, and 3.5-years post-infection using the 12-item Short Form Survey, Patient Health Questionnaire-9, Generalized Anxiety Disorder-7 scale and the Impact of Events Scale-Revised questionnaire. Additionally, a clinical symptom review was conducted for patients with persistent Long Covid at the 3.5-year follow-up. We had 149 respondents at 3 months, 94 at 1 year and 85 at 3.5-year. Of those who participated, 72% had Long Covid at the 3-month follow-up, with 26% and 25% having persistence of Long Covid symptoms at 1-year and 3.5 years, respectively. The most reported symptoms at the 3.5-year timepoint included fatigue, difficulty sleeping and easy crashing following activities. Overall, patients' Physical Composite Scores significantly improved between the 3-month and 3.5-year timepoints. However, the Physical Composite Scores of patients with persistent Long Covid were significantly lower than those of non-Long Covid patients at both the 3-month and 1-year follow-ups. The Mental Composite Score of persistent Long Covid patients remained significantly lower than individuals without Long Covid at all timepoints. At 3 months, Long Covid disproportionately met the criteria for depression, anxiety and PTSD symptoms. At 1 and 3.5 years, patients with persistent Long Covid were more likely to meet the criteria for depressive symptoms than those without Long Covid. Between the 3-months and 3.5-year timepoints, there was a significant reduction in the number of patients with persistent Long Covid who met the criteria for PTSD and anxiety symptoms. Although patients with Long Covid for 3.5 years had shown improvements in both their physical and mental health over time, they continue to lag behind those without Long Covid.

**Data availability statement:** All data used for this publication including the anonymized patient information, survey responses and Long Covid symptom review information are available from the Zenodo database (https://doi.org/10.5281/zenodo.14261918)

**Funding:** This work was supported by the Health Research Board (HRB) [COV19-2020-123]. In addition, the 2-4 year follow-up was supported by a grant from the Mater Foundation (23PAC104). The funders had no role in study design, data collection and analysis, decision to publish, or preparation of the manuscript

**Competing interests:** The authors have declared that no competing interests exist.

## Introduction

The COVID-19 pandemic caused over 770 million cases and 7 million COVID-19-related deaths reported to the WHO [1]. Global vaccination efforts have been an important contributor to reducing disease burden globally. The currently approved COVID-19 vaccines have proven effective in preventing severe disease [2]. Of those infected with COVID-19, only some go on experience persistent symptoms. This phenomenon was reported early in the pandemic in the spring of 2020 and the term "Long Covid" was coined [3]. According to the WHO, Long Covid is defined as signs or symptoms that persist or newly appear after an infection with COVID-19 [4]. Individuals may experience persistent symptoms or develop new ones three months after their infection, with symptoms lasting for at least two months and not explained an alternative diagnosis [4]. To date, studies have associated numerous symptoms with Long Covid. One Long Covid study identified over 200 potential symptoms associated with the condition [5]. However, many of these symptoms are non-specific and reported in observational studies that lack a control comparison group [6].

Early estimates of the prevalence of Long Covid from the WHO in 2022 were between 10–20% at 3 months post-infection [4]. However, a more recent study comparing 98,666 COVID-19 infected patients to uninfected matched controls in Scotland suggests that the prevalence is likely under 10% [6]. If confounding variables are adjusted for, the attributable prevalence of Long Covid is estimated to be around 6.6%, 6.5% and 10.4% at 6, 12 and 18 months respectively [6]. The underlying mechanisms responsible for Long Covid are complex and multifactorial. Several hypotheses have been proposed to explain the pathophysiology, including latent virus reactivation, microbiome disruption, endothelial dysfunction and clotting, immune dysregulation, neurological signaling dysfunction and autoimmunity [5]. Herpesvirus reactivation is a phenomenon that has been observed in the cohort of patients used in our study (Anticipate Cohort) [7]. These viral reactivations were primarily detected in patients requiring critical care and were associated with invasive ventilation and longer ventilation times [7].

While much research attention has focused on the physical symptoms of Long Covid, its psychological effects have been shown to have a significant burden on patients. During the acute phase of a COVID-19 infection, studies have identified elevated rates of mental health conditions, including anxiety, depression and post-traumatic stress disorder (PTSD). These have been linked to factors including inflammation [8], hospitalization [9], psychosocial stressors [10] and SARS-COV-2's neuroinvasive capability [11]. As many as one in three individuals infected with COVID-19 received a neurological or psychiatric diagnosis within 6 months of their infections [12]. These findings suggest that mental health complications can arise shortly after infection. In the context of patients living with Long Covid, these individuals generally experienced worse mental health outcomes than those without Long Covid. For patients living with Long Covid, symptoms such as fatigue, shortness of breath and chest tightness have been strongly associated with mental health conditions, including depression and anxiety [13]. Studies investigating mental health disorders in patients living with Long Covid have found that these individuals have

higher rates of anxiety, depression and PTSD than the general population [14–16]. Reported rates of these mental health conditions are as high as 31% for anxiety, 25% for depression and 29% for PTSD [15,16]. Compared to individuals who had COVID-19 but did not develop Long Covid, these patients tend to experience a greater number and more severe symptoms of depression [17]. Although rates of anxiety symptoms were similar between both groups, patients without Long Covid generally experienced better psychological recovery [17]. While rates of mental health conditions remain higher in patients with Long Covid up to two years post-infection, longitudinal studies have found that the severity of their symptoms slowly declined over time [15,18].

Of the studies investigating Long Covid, there are very few that extend beyond 18 months. At the time of writing, three published studies have investigated Long Covid patients two years after their original COVID-19 infections [19−21]. One study found that the most prevalent self-reported symptoms of Long Covid were fatigue, amnesia, concentration difficulty, insomnia and depression [21]. The second study reported that regardless of initial disease severity, the physical and mental health of individuals with Long Covid improved over a two-year period [19]. Despite the improvement in overall health, Long Covid patients' health remained lower than that of the general population and their symptom burden remained high [19]. The aim of our study is to further our understanding of Long Covid's impact on patients' physical and mental health 2–4 years after their initial infection, using the 12-item Short Form Questionnaire (SF-12). Additionally, we aim to characterize the persistent Long Covid symptoms and assess patients' mental health symptom burden using screening questionnaires for depression, anxiety and PTSD 3.5 years after initial infection. The current study is one of the first to investigate patients living with Long Covid over a 3.5-year period. This study is a follow-up to a previously published 1-year longitudinal study conducted with the same cohort [22].

## Materials and methods

### Patient recruitment and consent

Participants were recruited from the single-center prospective observational cohort previously investigated by O'Kelly, Vidal [22]. Patients were originally recruited from both inpatient and outpatient departments at the Mater Misericordiae University Hospital (MMUH) in Dublin, Ireland, between April 28, 2020 and October 30, 2020. At the initiation of the study protocol, patient information sheets and informed consent forms were provided to the subjects. The documents were reviewed and approved by a properly constituted Institutional Review Board/Independent Ethics Committee (IEC/IRB). Data protection was handled in compliance with national laws.

The Principal Investigator and/or designee explained to each patient the nature of the study, its purpose, the procedures involved, the expected duration and the potential risks and benefits involved. Each patient was informed that participation in the study was voluntary, that they could withdraw from the study at any time and that withdrawal of consent will not affect subsequent medical treatment or the relationship with the treating physician.

Two original written Patient Information and Consent Forms were completed, dated and personally signed by the patient and by the person responsible for collecting the informed consent. A third consent was additionally acquired at the final 3.5-year follow-up. In most cases, written consent was acquired from patients at the time the final questionnaire was completed. In cases where subjects did not return an up-to-date consent form for the 3.5-year follow-up, patients verbal consent for use of their information was acquired over the phone by the person responsible for collecting the informed consent. In these cases, the date, signature of the consenter and verbal consent were noted on the consent form.

### Definition of Long Covid and persistent Long Covid within the study

Identification of patients with Long Covid was conducted at the initiation of the study. To be considered as a patient with Long Covid, patients required a formal diagnosis by a qualified physician. Individuals were classified as having persistent Long Covid if they were diagnosed with Long Covid at the initiation of the study and reported ongoing symptoms at subsequent follow-ups. If a patient stopped reporting symptoms at any follow-up timepoint (i.e., 1 year or 3.5 years), they were removed from the

persistent Long Covid group for that and all future follow-ups. Only patients with persistent Long Covid at the 3.5-year timepoint were included in the symptom review. Patients without Long Covid or those who had previously had Long Covid but reported a cessation of symptoms were categorized as part of the no Long Covid group in the respective timepoints for analyses.

## Study design

Participants in our study were followed up at 3 months, 1 year and 2–4 years following their initial COVID-19 infection. All participants were contacted and provided the opportunity to respond to the study questionnaire at all three timepoints. This questionnaire consisted of subsections containing the SF-12, the Patient Health Questionnaire-9 (PHQ-9), the Generalized Anxiety Disorder-7 (GAD7) and the Impact of Events Scale-Revised (IESR) screening questions. The results of the 12-Item Short Form Survey (SF-12), and the persistence of Long Covid symptoms at the 3-month and 1-year follow-ups have been previously described by O'Kelly, Vidal [22]. Using the pre-existing data gathered during the aforementioned study, we followed up with the Anticipate Cohort 2–4 years after the initial infection to reassess the health status of those patients. Patients were followed up by phone or mail using the same questionnaire from O'Kelly, Vidal [22] to assess both their risk of mental health disorders and their overall physical and mental health. Patients who had persistent Long Covid symptoms at their most recent follow up in the O'Kelly, Vidal [22] study underwent a Long Covid clinical symptoms review. Patients were additionally queried about commonly reported Long Covid symptoms based on two meta-analyses that compared Long Covid patients to matched controls [19,21], and expert input from the Long Covid clinic doctors.

Symptoms of mental health disorders were examined using the PHQ-9, the GAD7 and IESR screening questionnaires. Questionnaires were previously administered at the 3-month and 1-year timepoints but were not previously reported on. The PHQ-9 [23] was used as a tool to screen for symptoms of major depression. The PHQ-9 is scored from 0–27 and a score of ≥5 was used as a cut-off for symptoms of major depression [24,25]. General anxiety disorder (GAD) symptoms were scored using the GAD7, which is scored from 0–21 [26]. We utilized a cut-off of ≥5 for the presence of GAD [26–28]. Lastly, presence of PTSD symptoms was assessed using the IESR, with a score ranging from 0–88 and a diagnostic cut-off score of 33 or greater [29]. The SF-12 was additionally used, which was previously reported on in O'Kelly, Vidal [22] comparing patients at 3 months and 1 year after their COVID-19 infection. The SF-12 survey is a shortened form of the SF-36 survey [30] which can be used to calculate the Mental Composite Score 12 (MCS12) and Physical Composite Score 12 (PCS12). Both the MCS and PCS scores are scored on a scale from 0–100 with a mean of 50 and a standard deviation of 10, assuming a standard population. An increase or decrease in the score reflects an above or below average score, respectively.

## Data analysis

Statistical analyses and plots were produced in RStudio V2022.12.0.353 [31]. The PCS12 and MCS12 subscores of the SF-12 questionnaire were calculated using compiled R code from Ottoboni et al. [32]. Plots were created using the RStudio package ggplot2 V3.4.4 [33]. Categorical data were expressed as absolute numbers and percentages. Categorical comparisons were statistically analysed using Chi-square tests or Fisher's exact test when appropriate. The normality of numerical variables was verified using the Shapiro-Wilk test. Data that were not normally distributed were compared using a Mann-Whitney U test or Wilcoxon signed-rank test to compare unpaired and paired variables respectively. Normally distributed data were statistically compared with paired and unpaired Student's t-tests. For all comparisons, incomplete questionnaires were filtered out and removed from the analyses.

## Results

### Demographics at timepoints and persistent Long Covid symptoms

From the Anticipate Cohort, a total of 149, 94 and 85 patients were included in this longitudinal study with follow-up at 3-months, 1 year and 3.5 years, respectively. The median response times at the three timepoints were 95 days (IQR

77.5–116) for the 3-month timepoint, 380 days (IQR 317–414) for the 1-year timepoint and 1238 days (IQR 1218.25–1315.75) for the 3.5-year timepoint. Patients were excluded from individual analyses if their questionnaires were incomplete. The demographics and comorbidities at initial presentation are presented in Table 1. The distribution of patients' acute COVID-19 symptoms at time of their initial COVID-19 infection is listed in S1 Table. The demographics of the patients captured at the three timepoints were proportionally similar (Table 1). The number of patients that reported persistent Long Covid symptoms decreased at the 1-year timepoint but remained unchanged between the 1-year and 3.5-year timepoints (Table 1). At the 3.5-year timepoint, six patients who had previously been diagnosed with persistent Long Covid during their 1-year or 3-month follow-ups did not respond to our inquiry about ongoing persistent Long Covid symptoms. However, these patients did complete the physical and mental health questionnaire for the 2–4-year follow-up. As a result, they were included in the corresponding sub-analyses but were excluded from the analyses related to persistent Long Covid.

The proportion of individuals from the Anticipate Cohort that responded to at least one of the three timepoints was predominantly female (68–69%), had comorbidities (70–79%) and 39–42% of patients had been hospitalized during their infection (Table 1). At the 3-month timepoint, 72% of the cohort had persistent Long Covid. This number decreased to 26%

**Table 1. Demographic information of patients who responded to questionnaire at 3-months, 1 year and 3.5 years.**

| Demographic | 3 month | | 1 year | | 3.5 years | |
|---|---|---|---|---|---|---|
| | N (%) | 95%CI | N (%) | 95%CI | N (%) | 95%CI |
| Total | 149 | | 94 | | 85 | |
| Female | 101 (68%) | 60-75% | 65 (69%) | 60-78% | 58 (68%) | 59-79% |
| Male | 48 (32%) | 25-39% | 28 (30%) | 21-39% | 27 (32%) | 22-42% |
| Hospital admission | 63 (42%) | 34-50% | 37 (39%) | 29-49% | 36 (42%) | 32-53% |
| ICU admission | 9 (6%) | 2-10% | 7 (7%) | 2-12% | 6 (7%) | 2-12% |
| Hospital readmission required | 15 (10%) | 5-15% | 11 (12%) | 5-19% | 8 (9%) | 3-15% |
| persistent Long Covid symptoms | 107 (72%) | 65-79% | 24 (26%) | 17-34% | 21 (25%) | 16-34% |
| Lost to follow-up for symptom review | 0 (0%) | | 0 (0%) | | 6 (7%) | 2-12% |
| Patients with comorbidities | 105 (70%) | 63-77% | 74 (79%) | 71-87% | 65 (76%) | 67-85% |
| **Comorbidities** | **3 month** | | **1 year** | | **3.5 years** | |
| | N (%) | 95%CI | N (%) | 95%CI | N (%) | 95%CI |
| Total | 105 | | 74 | | 65 | |
| Other pre-existing condition(s) | 98 (93%) | 88-98% | 72 (97%) | 93-100% | 62 (95%) | 90-100% |
| Asthma (requiring medication) | 14 (13%) | 7-19% | 10 (14%) | 6-22% | 10 (15%) | 6-24% |
| Diabetes | 12 (11%) | 5-17% | 7 (9%) | 2-16% | 6 (9%) | 2-16% |
| Cancer | 6 (6%) | 1-11% | 4 (5%) | 0-10% | 3 (5%) | 0-10% |
| Heart disease | 10 (10%) | 4-16% | 6 (8%) | 2-14% | 2 (3%) | 0-7% |
| Chronic haematological disorder | 2 (2%) | 0-5% | 2 (3%) | 0-7% | 2 (3%) | 0-7% |
| Chronic neurological impairment/disease | 4 (4%) | 0-8% | 2 (3%) | 0-7% | 2 (3%) | 0-7% |
| Chronic lung disease (non-asthma) | 3 (3%) | 0-6% | 2 (3%) | 0-7% | 1 (2%) | 0-5% |
| Pregnancy | 1 (1%) | 0-3% | 0 (0%) | | 0 (0%) | |
| HIV/other immune deficiency | 0 (0%) | | 0 (0%) | | 0 (0%) | |
| Chronic liver disease | 0 (0%) | | 0 (0%) | | 0 (0%) | |
| Chronic kidney disease | 1 (1%) | 0-3% | 1 (1%) | 0-3% | 0 (0%) | |
| Organ or bone marrow recipient | 0 (0%) | | 0 (0%) | | 0 (0%) | |

N = number of patients.

CI = Confidence interval.

at the 1-year timepoint and 25% at the 3.5-year timepoint (Table 1). The 21 patients who reported ongoing persistent Long Covid symptoms for 3.5 years reported the following symptoms: fatigue (76%), difficulty sleeping (76%), easy crashing following activities (76%), difficulty concentrating (62%), memory problems (52%), muscle pain (48%), joint pain (48%), hair loss (48%), palpitation/tachycardia (43%), anxiety (43%), shortness of breath (43%), numbness/tingling (43%), headaches (38%), loss of taste/smell (24%), confusion (24%), chest pain (24%), unusual sweating (19%), ear problems (19%), stomach pain (14%), dizziness (10%), cough (10%), diarrhoea (10%), decrease muscle strength (10%), change in voice (5%), constipation (5%), vomiting regularly (5%), persistent sore throat (5%), neuralgia (5%) and vision problems (5%). The most common symptoms included fatigue-related symptoms with fatigue, difficulty with sleep and easy crashing following activities being the most common (Table 1). Neurological manifestations (difficulty concentrating, memory problems, numbness/tingling, confusion, dizziness), cardiovascular symptoms (palpitation/tachycardia) and respiratory symptoms (shortness of breath, chest pain, cough) were also among the more common symptoms reported by Long Covid patients. The number of symptoms reported by patients ranged from 1 to 18 persistent symptoms, with a median of 9 (IQR 4.5–12).

## Factors associated with Physical PCS12 and MCS12 score changes over time

Using the SF-12 questionnaire, we calculated the MCS12 and Physical composite PCS12 scores to compare patients at the 3.5-year timepoint compared to the earlier, previously published 3-month timepoint [22]. Paired comparisons of all patients and subgroups were normally distributed (S2 Table). In all comparisons, the MCS12 scores did not significantly change between the 3-month and 3.5-year timepoints (S1 Fig). However, the PCS12 scores significantly increased over time overall in the cohort and in all subgroups (Fig 1). This data demonstrated that over time, the mental health of patients did not change between 3 months and 3.5 years. However, the physical health of the overall cohort did improve.

The MCS12 and PCS12 scores of the subgroups were further investigated using the data of all respondents at the respective timepoints. In all subgroups, the data were not normally distributed (S3 Table) and were statistically compared using the Mann-Whitney U test. MCS12 scores of patients with persistent Long Covid were significantly worse at the 3-month and 1-year follow-ups compared to patients who did not have Long Covid (Fig 2). Patients with comorbidities and those who required hospital readmission additionally had lower MCS12 scores at 1 year following their infections (Fig 2). As for the PCS12 scores, persistent Long Covid patients had lower scores at all timepoints (Fig 3). Additionally, patients with comorbidities had lower PCS12 scores at the 1-year and 3.5-year follow-ups (Fig 3).

## Depression, anxiety and PTSD symptom scores over time

Patients who met the criteria for depression, anxiety or PTSD symptoms using the PHQ-9 (score ≥ 5), GAD7 (score ≥ 5) and IESR (score ≥33) questionnaires, respectively, are shown in Table 2. Within the Anticipate Cohort there were significantly more patients with persistent Long Covid who met the criteria for depressive symptoms at the 3-month, 1-year and 3.5-year timepoints compared to non-Long Covid patients. Patients with persistent Long Covid also met the symptom cutoff for PTSD and anxiety at the 3-month timepoint, but not at the later timepoints (Table 2). When comparing the initial 3-month timepoint to the later timepoints for all patients, we observed a significant decrease in the number of individuals who met the criteria for PTSD symptoms between 3-month and the 3.5-year timepoint. Subgroup analyses of the patients revealed significant decreases in the proportion of patients meeting the PTSD symptom cutoff across several groups, including patients with comorbidities, those who did not require hospitalization, ICU care, or readmission to hospital and those with persistent Long Covid (Table 2). In the case of anxiety symptoms, patients with persistent Long Covid were the only group to demonstrate a significant decrease at the 3.5-year timepoint compared to the initial 3-month timepoint. There was no significant change in the proportion of persistent Long Covid patients meeting the PHQ-9 cutoff for depressive symptoms between the 3-month and 3.5-year timepoints (Table 2). However, a significant decrease was observed among patients without Long Covid. At the 1-year and 3.5-year timepoints, depressive symptoms were the only mental health symptoms that were disproportionately present in persistent Long Covid patients compared to individuals without

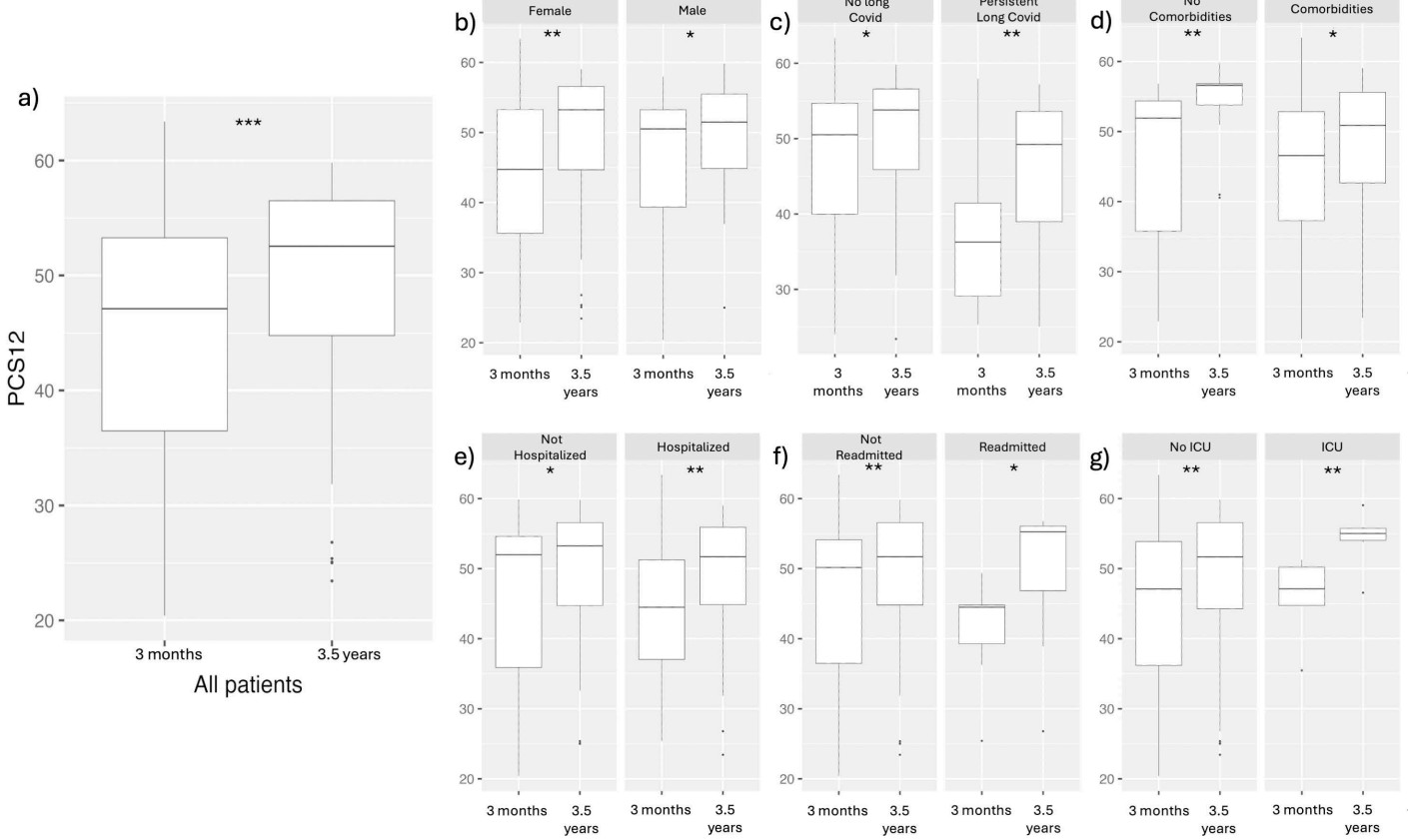

**Fig 1. PCS12 paired score comparison of patients 3 months and 3.5 years.** (A) Comparison of all patients. Subgroups analyses comparing patients by (B) sex, (C) persistent Long Covid status, (D) presence of comorbidities, (E) hospitalization status, (F) readmission to hospital and (G) ICU care requirement on admission. P-value < 0.05 (*), P-value < 0.01 (**), P-value < 0.001 (***).

Long Covid (Table 2). No mental health symptoms were disproportionately present in any other subgroups at the 1-year timepoint (Table 2). However, at the 3.5-year timepoint, IESR scores greater than or equal to 33 were more prevalent among patients who had been hospitalized or required a hospital readmission compared to their non-hospitalized or non-readmitted counterparts.

## Discussion

The aim of this study was to analyse results from one of the longest-running cohort studies on persistent Long Covid patients, building on the previously published 1-year follow-up study by O'Kelly, Vidal [22]. Our data suggest that 3.5-years after their original infection, most patients who had persistent Long Covid at their 1-year follow-up continued to have symptoms at the 3.5-year follow-up. Similar results have been reported in published longitudinal 2-year studies, where patients continue to report persistence of symptoms with no significant improvement over time [19,21]. Fatigue-related symptoms were the most commonly reported by patients at the 3.5-year timepoint, and gastrointestinal symptoms were the least commonly reported. These findings are consistent with the aforementioned 2-year follow-up studies [19,21]. Although our cohort is small, we captured a wide range of symptoms that have continued to affect patients. In many cases, patients were burdened by multiple symptoms. The median number of symptoms reported by the cohort was 9, with a maximum of 18 symptoms reported by a single individual.

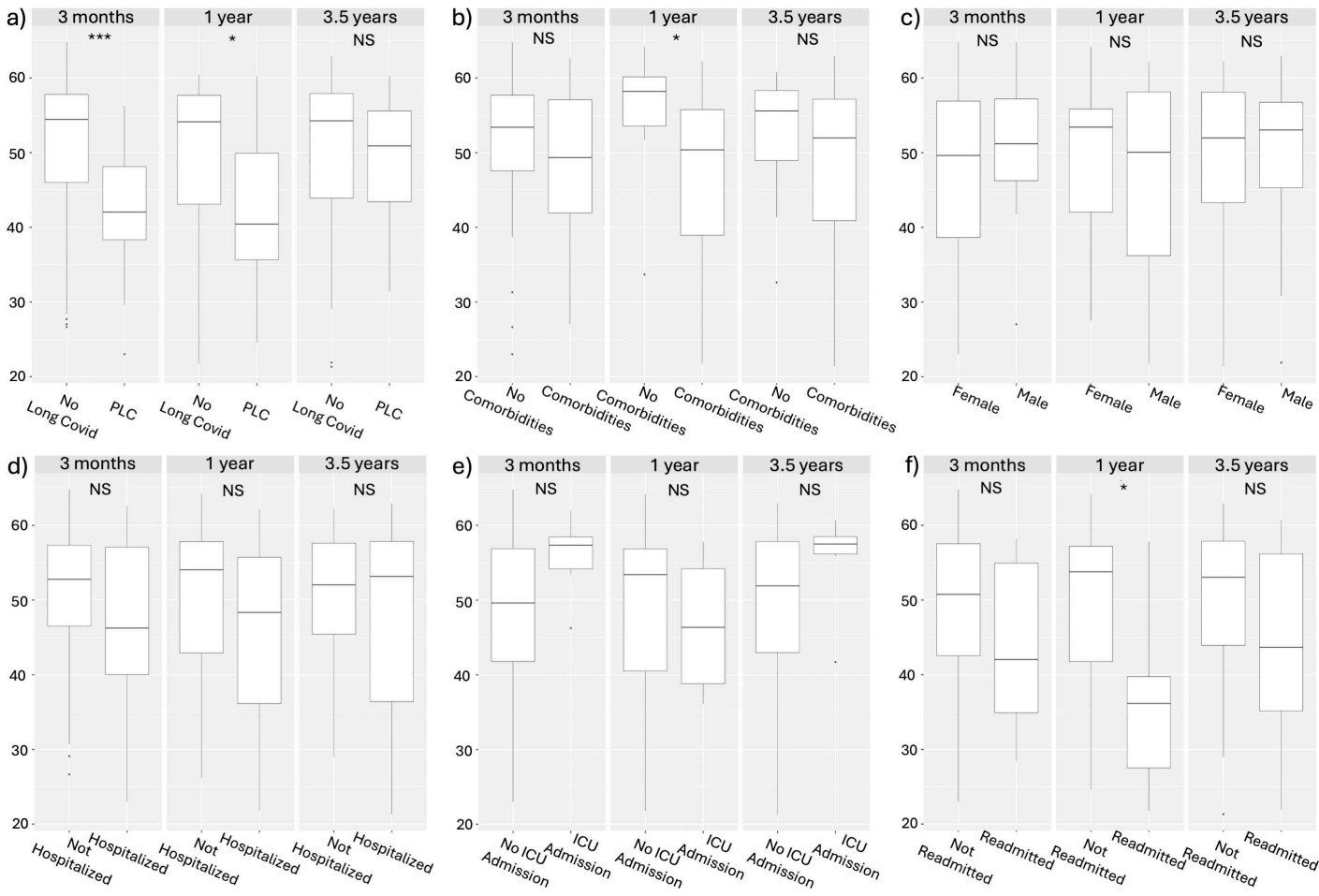

**Fig 2. MCS12 score comparison of sub-populations at 3 months, 1 year and 3.5 years.** Comparison of patients based on (A) persistent Long Covid (PLC) status, (B) presence of comorbidities, (C) sex, (D) hospitalization status, (E) ICU care requirement on admission and (F) readmission to hospital. P-value < 0.05 (*), P-value< 0.01 (**), P-value< 0.001 (***).

Using the SF-12 questionnaire, we assessed patients' general mental health and physical health using the MCS12 and PCS12 scores. We found that mental health scores of the overall group, as well as within each subgroup, did not significantly change between the 3-month and 3.5-year timepoints (S1 Fig). These results are similar to our previous 1-year follow-up study [22], where we did not observe a significant change in the MCS12 score when comparing the 1-year scores to the 3-month timepoint scores. Despite no significant changes in the MCS12 over time, subgroup comparisons did reveal some significant differences between subgroups at different timepoints (Fig. 2). Of note, the persistent Long Covid patients' MCS12 scores were significantly lower than those of patients without Long Covid at the 3-month and 1-year timepoints. This suggests that, at early timepoints of our study, patients living with persistent Long Covid had worse general mental health than those without Long Covid. Alongside MCS12 scores, we observed improvements in the PCS12 scores of patients in the study. The PCS12 scores improved between the 3-month and 3.5-year timepoints (Fig 1). This was both true across the  cohort as a whole and within all subgroups, indicating that all patients' physical health improved over time. Although improvements were observed, the PCS12 scores for those living with persistent Long Covid for 3.5 years remained significantly lower than individuals without Long Covid at all three timepoints (Fig 3). This

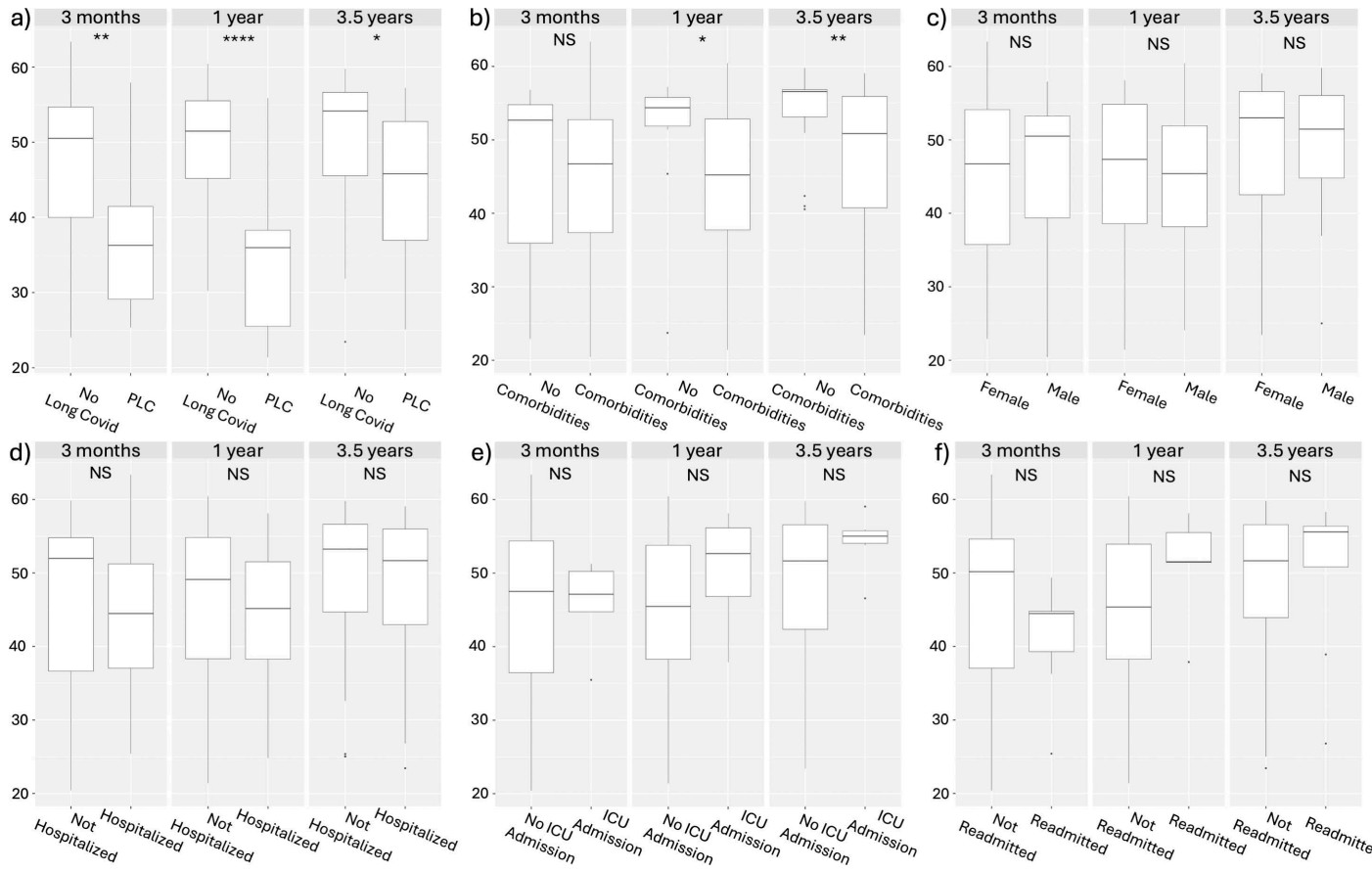

**Fig 3. PCS12 score comparison of sub-populations at 3 months, 1 year and 3.5 years.** Comparison of persistent Long Covid (PLC) status, (B) presence of comorbidities, (C) sex, (D) hospitalization status, (E) ICU care requirement on admission and (F) readmission to hospital. P-value < 0.05 (*), P-value < 0.01 (**), P-value < 0.001 (***), P-value < 0.0001 (****).

highlights that, despite improvements in the health of patients living with persistent Long Covid, their physical health continues to remain well below that of those who have fully recovered from their Covid infections. Similar physical and mental health results were demonstrated in a 2-year follow-up study, showing long-term improvement in mental and physical health using a health-related quality of life questionnaire [19]. The 2-year follow-up study also demonstrated that, despite improvement, patients with Long Covid had lower health status when compared to patients who did not have Long Covid [19].

Despite no significant change in the MCS12 scores of persistent Long Covid patients between the 3-month and 3.5-years timepoints, we observed significant changes in the mental health questionnaires. Specifically, we found that significantly fewer persistent Long Covid patients met the thresholds for anxiety (GAD7 < 5) and PTSD (IESR < 33) symptoms when comparing the 3-month and 3.5-year timepoints (Table 2). No significant improvements in PTSD or GAD7 scores were apparent between the 3-month and 1-year timepoints (Table 2). These results are similar to those observed by Huang, Li [19] in terms of the lack of significant improvement in anxiety symptoms at 1 year following infection compared to their initial 6-month follow-ups. They observed a significant decrease in the number of Long Covid patients with anxiety symptoms 2 years after their infection [19]. In contrast to Huang, Li [19], who observed significant decreases in depression symptoms in Long Covid patients from their 6 month to 2 year follow-up, we did not observe significant changes in

Table 2. PHQ-9, GAD7 and IESR Score of cohort subgroups at 3 months, 1 year and 3.5 years.

| Comparisons | | PHQ-9 (≥5) Timepoint | | | GAD7 (≥5) Timepoint | | | IESR (≥33) Timepoint | | |
|---|---|---|---|---|---|---|---|---|---|---|
| | | 3 months | 1 year | 3.5 years | 3 months | 1 year | 3.5 years | 3 months | 1 year | 3.5 years |
| all patients | | 71/139(43) | 44/92(47) | 34/82(41) | 56/147(39) | 30/93(32) | 21/83(25) | 21/110(19) | 14/80(18) | 5/79(6)^A |
| Persistent Long Covid at the 3.5 year timepoint | Persistent Long Covid | 16/19(84) | 10/13(77) | 12/19(63) | 14/20(70) | 8/13(62) | 6/20(30)^A | 7/15(47) | 4/11(36) | 1/19(5)^A |
| | No Long Covid | 53/113(47) | 31/74(42) | 18/57(32)^A | 41/121(34) | 22/75(29) | 13/57(23) | 13/91(14) | 10/66(15) | 4/55(7) |
| | Statistics within timepoints (p-value) | **0.003** | **0.020** | **0.015** | **0.002** | 0.053^B | 0.555^B | **0.008**^B | 0.107^B | 1^B |
| Hospitalization status at initial infection | Hospitalized | 34/59(58) | 17/36(47) | 15/34(44) | 28/86(33) | 17/36(47) | 9/35(26) | 10/41(24) | 8/29(28) | 5/31(16) |
| | Not hospitalized | 36/78(46) | 26/55(47) | 18/47(38) | 27/60(45) | 13/56(23) | 12/47(26) | 11/69(16) | 6/50(12) | 0/47(0)^A |
| | Statistics within timepoints (p-value) | 0.183 | 0.996 | 0.599 | 0.127 | **0.017** | 0.985 | 0.276 | 0.080 | **0.008**^B |
| Re-admitted at initial infection | Re-admitted to hospital | 11/15(73) | 6/10(60) | 4/7(57) | 10/14(71) | 6/10(60) | 4/8(50) | 5/12(42) | 3/8(38) | 3/7(43) |
| | Not readmitted | 59/122(48) | 37/81(46) | 29/74(39) | 45/132(34) | 24/82(29) | 17/74(23) | 16/98(17) | 11/71(16) | 2/71(3)^A |
| | Statistics within timepoints (p-value) | 0.068 | 0.508^B | 0.435^B | **0.006** | 0.073^B | 0.194^B | 0.050^B | 0.145^B | **0.004**^B |
| Comorbidities present | Comorbidities | 56/99(57) | 37/73(51) | 26/61(43) | 39/102(38) | 26/73(36) | 18/63(29) | 16/73(22) | 12/63(19) | 5/59(8)^A |
| | No comorbidities | 14/38(37) | 6/18(33) | 7/20(35) | 16/44(36) | 4/19(21) | 3/19(16) | 5/37(14) | 2/16(13) | 0/19(0) |
| | Statistics within timepoints (p-value) | 0.387 | 0.187 | 0.547 | 0.830 | 0.228 | 0.373^B | 0.289 | 0.723^B | 0.327^B |
| ICU admission at initial infection | ICU admission | 3/9(33) | 2/7(29) | 1/6(17) | 4/9(44) | 3/7(43) | 0/6(0) | 0/6(0) | 1/6(17) | 0/6(0) |
| | No ICU | 67/128(52) | 41/84(49) | 32/75(43) | 51/137(37) | 27/84(32) | 21/76(28) | 21/104(20) | 13/73(18) | 5/72(7)^A |
| | Statistics within timepoints (p-value) | 0.319^B | 0.440^B | 0.393^B | 0.729^B | 0.679^B | 0.330^B | 0.593^B | 1^B | 1^B |
| Sex | Male | 20/45(44) | 12/38(32) | 10/26(38) | 14/46(30) | 11/28(39) | 4/25(16) | 6/35(17) | 6/25(24) | 1/25(4) |
| | Female | 50/92(54) | 31/63(49) | 23/55(41) | 41/100(41) | 19/64(30) | 17/57(30) | 15/75(20) | 8/54(15) | 4/53(8) |
| | Statistics within timepoints (p-value) | 0.276 | 0.576 | 0.774 | 0.221 | 0.366 | 0.187 | 0.723 | 0.353^B | 1^B |

^A Statistically significant difference between the 3 month and 3.5-year timepoints (p<0.05). See S4 table for exact p-values.

^B Fisher's exact test was used instead of Chi-square test because at least one of the expected cell counts was less than 5 in the statistical comparison.

depressive symptoms (PHQ-9) scores over time for patients living with persistent Long Covid. Although we observe a 21% decrease in patients with a PHQ-9 <5, the lack of statistical significance can likely be attributed to our small cohort size. It is well documented through self-reported and questionnaire-based approaches that Long Covid patients struggle with anxiety and depression symptoms [19,21]. We also found that patients with persistent Long Covid disproportionately met criteria for symptoms of depression at all three timepoints compared to patients who did not have Long Covid, consistent with previous studies. In contrast, anxiety and PTSD symptoms were only disproportionately present in the Long Covid patient subgroup at the 3-month timepoint.

Overall, our study demonstrates that patients continue to experience a variety of Long Covid symptoms 3.5 years after their original COVID-19 infection. The proportion of patients living with persistent Long Covid at the 3.5-year timepoint remains similar to the 1-year timepoint as supported by other longitudinal studies [19,21]. The symptom burden for these patients continues to be high. However, despite the high symptom burden, the physical and mental health of patients living with persistent Long Covid has improved. At 3.5 years post-infection, the mental health scores (MCS12) of patients with persistent Long Covid are similar to those without Long Covid. The proportion of patients living with persistent Long Covid meeting the criteria for PTSD symptoms has also improved to levels similar to those without Long Covid. Despite improvements in general mental health, a greater proportion of patients with persistent Long Covid still meeting the PHQ-9 criteria for depressive symptoms. This finding is consistent within our cohort of patients at all timepoints. Regarding the physical health of individuals living with persistent Long Covid, we found that their physical health has improved over time. However, it is important to recognize that their physical health continues to lag behind that of individuals without Long Covid.

While our findings indicate that a significantly higher proportion of individuals with persistent Long Covid had depressive symptoms at all three timepoints, it is important to note that we cannot infer causation from this study. Given that this is a prospective cohort study, we do not have baseline mental health questionnaire scores prior to patients' COVID-19 infections. We can simply infer that longitudinally, more patients with persistent Long Covid met the criteria for depressive symptoms when screened with the PHQ-9 compared to those without persistent symptoms. As the PHQ-9 is a screening tool and not a diagnostic instrument, it is important to understand that the results reflect symptom severity rather than clinical diagnosis. While our study supports the longitudinal association between Long Covid, depressive symptoms and its persistence over time [15,17,34], we caution against interpreting this as evidence of direct causation. Low mood is likely multifactorial involving both patients' persistent symptoms and other factors such as age, pre-existing psychiatric conditions, severity of the initial infection, socioeconomic factors, social isolation, employment difficulties and health concerns [13,15,35]. These findings suggest that depressive symptoms may reflect complex interactions between broader biopsychosocial factors and the psychological burden of living with a debilitating chronic illness. Although we are unable to infer causation from our data, it highlights the importance of integrating mental health screening and monitoring into long-term management strategies for individuals living with persistent Long Covid.

## Supporting information

**S1 Fig. MCS12 paired score comparison of patients 3 months and 3.5 years.** Comparison of all patients. Subgroups analyses comparing patients by sex, persistent Long Covid status, Presence of comorbidities, hospitalization status, hospital readmissions and ICU care requirement on admission.
(TIF)

**S1 Table. Acute symptoms reported by patients at initial presentation with COVID.** N = number of Patients. CI = Confidence interval.
(PDF)

**S2 Table. Shapiro-wilk test of normality for paired comparisons.** [A] Test was conducted on the difference between means of the paired data at the respective timepoints.

(PDF)

**S3 Table. Shapiro-wilk test of normality of unpaired subgroups at all timepoints.**
(PDF)

**S4 Table. P-values for mental health questionnaire statistical comparisons between timepoints in Table 2.** [A] The Fisher's exact test was used instead of Chi-square test because one of the expected cell values was less than 5 in the statistical comparison.
(PDF)

## Acknowledgments

The authors would like to thank the patients for their continued participation throughout this study.

## Author contributions

**Conceptualization:** Gordana Avramovic, Brendan O'Kelly, John S. Lambert.

**Data curation:** Gregory Vallée, David Xi, Brendan O'Kelly, John S. Lambert.

**Formal analysis:** Gregory Vallée, Brendan O'Kelly.

**Funding acquisition:** Gordana Avramovic, John S. Lambert.

**Investigation:** Gregory Vallée, David Xi, Brendan O'Kelly, John S. Lambert.

**Methodology:** Gregory Vallée.

**Project administration:** Gordana Avramovic, John S. Lambert.

**Resources:** John S. Lambert.

**Supervision:** Gordana Avramovic, John S. Lambert.

**Visualization:** Gregory Vallée.

**Writing – original draft:** Gregory Vallée.

**Writing – review & editing:** David Xi, Gordana Avramovic, Brendan O'Kelly, John S. Lambert.

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
