## [Decision Letter · Decision Letter 0]

19 Mar 2025

PONE-D-24-56255Evaluating the longitudinal physical and psychological health effects of Long Covid at 3.5 years post-infectionPLOS ONE

Dear Dr. Lambert,

Thank you for submitting your manuscript to PLOS ONE. After careful consideration, we feel that it has merit but does not fully meet PLOS ONE’s publication criteria as it currently stands. Therefore, we invite you to submit a revised version of the manuscript that addresses the points raised during the review process.

We look forward to receiving your revised manuscript.

Kind regards,

Vincenzo De Luca

Academic Editor

PLOS ONE

Journal Requirements:

“This work was supported by the Health Research Board (HRB) [COV19-2020-123]. In addition, the 2-4 year follow-up was supported by a grant from the Mater Foundation (23PAC104).”

Reviewers' comments:

Reviewer's Responses to Questions

**Comments to the Author**

1. Is the manuscript technically sound, and do the data support the conclusions?

Reviewer #1: Yes

Reviewer #2: Yes

2. Has the statistical analysis been performed appropriately and rigorously? 

Reviewer #1: Yes

Reviewer #2: Yes

3. Have the authors made all data underlying the findings in their manuscript fully available?

Reviewer #1: Yes

Reviewer #2: Yes

4. Is the manuscript presented in an intelligible fashion and written in standard English?

Reviewer #1: Yes

Reviewer #2: Yes

5. Review Comments to the Author

Reviewer #1: I found this to be an interesting an important contribution to our understanding of long Covid. The paper is well written, (though it needs some editing for typos etc) and mostly clear. I had two questions. First, do they have information as to why subjects dropped out of the study? Do they know if it was disinterest, death or possibly long Covid symptoms that reduced motivation? This might impact their results. Second, while perhaps it is obvious but wasn't clear from my reading, were the subjects with long Covid at 3 months, 1 yr and 3.5 yrs the same subjects? There was no late emergence of long Covid symptoms?

Reviewer #2: Review Comments

This study is an important and informative contribution to the field, particularly in its investigation of the long-term psychological impact of Long Covid. However, there are few areas that require further clarification and elaboration.

1. Background and Literature Review:

a. While the study aims to assess the mental health impact of Long Covid, including depression, anxiety, and post-traumatic stress disorder (PTSD), the background section does not provide sufficient discussion on the relationship between COVID-19 infection and these mental health outcomes.

b. The authors should include a review of existing literature that demonstrates how COVID-19 infection, particularly in its early stages, has been linked to depression, anxiety, and PTSD.

c. Additionally, evidence of the prevalence and severity of these mental health conditions at the time of acute infection, or shortly thereafter, should be incorporated into the background section to contextualize the study’s findings.

2. Causality of Depression in Long Covid Patients:

a. The results indicate that a significantly higher proportion of patients with persistent Long Covid met the criteria for depression at multiple timepoints (3 months, 1 year, and 3.5 years post-infection).

b. However, the study does not establish COVID-19 as the primary cause of depression. Other potential contributing factors—such as pre-existing mental health conditions, socioeconomic stressors, and pandemic-related disruptions—should be acknowledged in the discussion section.

c. The authors should also consider discussing whether other confounding factors, such as prolonged physical symptoms, social isolation, or employment difficulties, could have contributed to the persistence of depressive symptoms.

3. Interpretation of Findings:

a. The discussion should clarify whether the persistence of depressive symptoms in Long Covid patients represents a direct consequence of the viral infection, a secondary effect of chronic illness, or an interaction between multiple biopsychosocial factors.

b. If possible, the authors should compare their findings with other long-term studies on post-viral syndromes to strengthen their conclusions.

4. Clarity in Reporting:

a. The authors should also ensure consistency in terminology when referring to “Long Covid” and clarify how they distinguish persistent Long Covid from other post-infection conditions.

Conclusion

Overall, this study provides valuable insights into the long-term psychological health effects of Long Covid. However, addressing the points above—particularly strengthening the background section, discussing alternative explanations for depression would significantly enhance the manuscript’s clarity and scientific rigor.

6. PLOS authors have the option to publish the peer review history of their article (what does this mean? ). If published, this will include your full peer review and any attached files.

**Do you want your identity to be public for this peer review?** For information about this choice, including consent withdrawal, please see our Privacy Policy .

Reviewer #1: **Yes: ** William Sulis

Reviewer #2: **Yes: ** Dr Bala Isa Harri

---

## [Author Response · Author response to Decision Letter 1]

3 May 2025

Dear Academic editor and Reviewers,

We would like to begin by thanking you for providing us with your review and suggestions for our manuscript. We have attempted to address your suggestions and concerns within the manuscript. Additionally, we have addressed the comments you provided and answered any questions you sent us. We have have also proofread the document and corrected grammatical errors and any inconsistencies in acronyms or discrepancies in capitalizations (ex: Long Covid vs long Covid). We have provided a response to all your comments and queries below your original comments.

Academic editor:

Comment 1:

Response 1:

Thank you for providing the linked templates as a guide. We have adjusted the title page and the manuscript to meet the PLOS ONE style requirements.

Comment 2:

Please note that funding information should not appear in any section or other areas of your manuscript. We will only publish funding information present in the Funding Statement section of the online submission form. Please remove any funding-related text from the manuscript.

Response 2:

We have removed the statement in our acknowledgement section that refers to our funding sources.

Comment 3:

Thank you for stating the following financial disclosure:

“This work was supported by the Health Research Board (HRB) [COV19-2020-123]. In addition, the 2-4 year follow-up was supported by a grant from the Mater Foundation (23PAC104).”

Response 3:

We have submitted a new cover letter with our article with the amended statement including the role of the agencies that funded our work as you requested. Please see the cover letter for the amended statement.

Reviewer #1:

Comment:

I found this to be an interesting an important contribution to our understanding of long Covid. The paper is well written, (though it needs some editing for typos etc) and mostly clear. I had two questions. First, do they have information as to why subjects dropped out of the study? Do they know if it was disinterest, death or possibly long Covid symptoms that reduced motivation? This might impact their results. Second, while perhaps it is obvious but wasn't clear from my reading, were the subjects with long Covid at 3 months, 1 yr and 3.5 yrs the same subjects? There was no late emergence of long Covid symptoms?

Answer to “First, do they have information as to why subjects dropped out of the study? Do they know if it was disinterest, death or possibly long Covid symptoms that reduced motivation?”:

During both follow-ups periods (1-year and 3.5-year) of our study, we attempted on numerous occasions to contact the participants by phone and mail. In most cases, the patients lost to follow-up were due to the lack of response following multiple attempts of reaching the participant over the span of 2-3 months. In a few cases, participants were deceased or expressed that they did not wish to participate in the follow-up. The exact reason for which participants declined to participate were not noted.

Answer to “Second, while perhaps it is obvious but wasn't clear from my reading, were the subjects with long Covid at 3 months, 1 yr and 3.5 yrs the same subjects? There was no late emergence of long Covid symptoms?”

The subjects with Long Covid this study were individuals who had formal diagnoses of Long Covid at the initiation of the study (3-month time point). The final 3.5-year symptom reviews were conducted only on individuals who had been diagnosed with Long Covid and continued to report Long Covid symptoms at all subsequent follow-ups that the individual participated in. Therefore, patients who did not receive a diagnosis of Long Covid at the 3 month or no longer reported having Long Covid symptoms at the 1-year follow-up were not included in the final symptom review. This was completed in this manner to examine the longitudinal impact of Long Covid of patients living with the condition for 3.5-years. While we acknowledge that a late emergence case may be missed. Our primary aim was to investigate patients living with Long Covid for 3.5-years. We have changed the language in our paper to make this clearer as this was a similar concern that was brought up by reviewer #2.

Reviewer #2:

Comment 1:

Background and Literature Review:

a. While the study aims to assess the mental health impact of Long Covid, including depression, anxiety, and post-traumatic stress disorder (PTSD), the background section does not provide sufficient discussion on the relationship between COVID-19 infection and these mental health outcomes.

b. The authors should include a review of existing literature that demonstrates how COVID-19 infection, particularly in its early stages, has been linked to depression, anxiety, and PTSD.

c. Additionally, evidence of the prevalence and severity of these mental health conditions at the time of acute infection, or shortly thereafter, should be incorporated into the background section to contextualize the study’s findings.

Answer 1:

To address your concerns regarding the limited discussion on mental health outcomes associated with COVID and Long Covid in the introduction section, we have revised the introduction of the manuscript accordingly. We had added an additional paragraph to provide an overview of the existing literature linking COVID to depression, anxiety and PTSD. We have also incorporated relevant literature highlighting the prevalence and severity of these mental health conditions during or shortly after the acute phase of COVID infections and in individuals with Long Covid.

Comment 2:

Causality of Depression in Long Covid Patients:

a. The results indicate that a significantly higher proportion of patients with persistent Long Covid met the criteria for depression at multiple timepoints (3 months, 1 year, and 3.5 years post-infection).

b. However, the study does not establish COVID-19 as the primary cause of depression. Other potential contributing factors—such as pre-existing mental health conditions, socioeconomic stressors, and pandemic-related disruptions—should be acknowledged in the discussion section.

c. The authors should also consider discussing whether other confounding factors, such as prolonged physical symptoms, social isolation, or employment difficulties, could have contributed to the persistence of depressive symptoms

Answer 2:

We have clarified in the discussion of our study that the prospective cohort design used in this study does not allow us to infer causation between Long Covid and depression. We also emphasize that questionnaires like the PHQ-9 function as a screening tool that reflects symptomatology and is not inherently diagnostic. We have also revised the language in the paper to better reflect that our findings are about the presence of symptoms of mental health conditions and is not diagnostic of said mental health conditions. While individuals living with Long Covid have greater symptoms of depression longitudinally, we acknowledge that the presence and persistence of these symptoms are likely multifactorial and cite relevant literature that supports the multifactorial relationship between Long Covid and mental health. We have added a paragraph to the discussion to help address these concerns.

Comment 3:

Interpretation of Findings:

a. The discussion should clarify whether the persistence of depressive symptoms in Long Covid patients represents a direct consequence of the viral infection, a secondary effect of chronic illness, or an interaction between multiple biopsychosocial factors.

b. If possible, the authors should compare their findings with other long-term studies on post-viral syndromes to strengthen their conclusions.

Answer 3:

Much like the points in your second comment, we have emphasized that the depressive symptoms in individuals with persistent long covid are unlikely to be a result of a single factor. The cause of these symptoms is most likely multifactorial and is a combination of Long Covid and its interplay with biopsychosocial factors. We have elaborated on this point in the discussion section of our manuscript to address this overlap and cited relevant literature to support it.

Comment 4:

Clarity in Reporting:

c. The authors should also ensure consistency in terminology when referring to “Long Covid” and clarify how they distinguish persistent Long Covid from other post-infection conditions.

Answer 4:

Thank you for identifying the lack of precision in the terminology that we used in this study. We have added a subsection to the method section of the manuscript help describe our definition of Long Covid and Persistent Long Covid. Long Covid patients were identified in our study were defined as any individual who had received a formal diagnosis of Long Covid by a qualified physician at the initiation of our study. Those who qualified as having persistent Long Covid would have a formal diagnosis in addition to reporting ongoing Long Covid symptoms at all timepoints for which they participated in the study. If a patient no longer reported symptoms in an earlier timepoint, then they would not qualify as a persistent Long Covid patient in subsequent timepoints. Our studies primary focus was on individuals with persistent Long Covid Symptoms over the 3.5-year time period. We have adjusted the language in the manuscript in its entirety to better reflect that we are investigating patients who have been living with persistent Long Covid. We have also adjusted the title to better reflect this.

We would like to sincerely thank you for taking the time to read our article and provide valuable feedback. We hope that the additions we made and clarifications in the language we used in our manuscript will be to your liking.

We look forward to your response,

Gregory Vallée, David Xi, Gordana Avramovic, Brendan O’Kelly and John S. Lambert

---

## [Decision Letter · Decision Letter 1]

5 June 2025

Evaluating the longitudinal physical and psychological health effects of persistent Long Covid 3.5 years after infection

PONE-D-24-56255R1

Dear Dr. Lambert,

We’re pleased to inform you that your manuscript has been judged scientifically suitable for publication and will be formally accepted for publication once it meets all outstanding technical requirements.

Kind regards,

Vincenzo De Luca

Academic Editor

PLOS ONE

Additional Editor Comments (optional):

Reviewers' comments:

Reviewer's Responses to Questions

**Comments to the Author**

1. If the authors have adequately addressed your comments raised in a previous round of review and you feel that this manuscript is now acceptable for publication, you may indicate that here to bypass the “Comments to the Author” section, enter your conflict of interest statement in the “Confidential to Editor” section, and submit your "Accept" recommendation.

Reviewer #1: All comments have been addressed

2. Is the manuscript technically sound, and do the data support the conclusions?

Reviewer #1: Yes

3. Has the statistical analysis been performed appropriately and rigorously? 

Reviewer #1: Yes

4. Have the authors made all data underlying the findings in their manuscript fully available?

Reviewer #1: Yes

5. Is the manuscript presented in an intelligible fashion and written in standard English?

Reviewer #1: Yes

6. Review Comments to the Author

Reviewer #1: (No Response)

7. PLOS authors have the option to publish the peer review history of their article (what does this mean? ). If published, this will include your full peer review and any attached files.

**Do you want your identity to be public for this peer review?** For information about this choice, including consent withdrawal, please see our Privacy Policy .

Reviewer #1: No

---

## [Editor Report · Acceptance letter]

PONE-D-24-56255R1

PLOS ONE

Dear Dr. Lambert,

I'm pleased to inform you that your manuscript has been deemed suitable for publication in PLOS ONE. Congratulations! Your manuscript is now being handed over to our production team.

Kind regards,

on behalf of

Dr. Vincenzo De Luca

Academic Editor

PLOS ONE